# Selection of Maintenance Strategies for Machines in a Series-Parallel System

**Bożena Zwolińska and Jakub Wiercioch ***

Faculty of Mechanical Engineering and Robotics, AGH University of Science and Technology, 30-059 Kraków, Poland
* Correspondence: wiercioc@agh.edu.pl; Tel.: +48-607-940-770

**Abstract:** In this paper, an assessment of the failure frequency of machines in a series-parallel structure was conducted. The analyses contain the decomposition of the system according to the general theory of complex systems. Based on the results obtained, a model for an optimal determination of the mean time to failure (MTTF) according to the expected value of the gamma distribution was proposed. For this purpose, the method of moments was used to determine the optimal values of the parameters of the estimated gamma distribution. The article is designed to be analytical. The object of consideration in this analysis is the real production system working in accordance with make-to-order, with a high degree of product customisation. Moreover, in the considered system occurs a dichotomy of mutually exclusive flows: push and pull. In the article, the main emphasis was placed on the applicability of the proposed MTTF value-shaping algorithm. Then, the maintenance strategy for each machine (reactive, preventive or predictive) was proposed. Maintenance strategy selection considered sustainable development principles in the criterion of minimizing maintenance actions, fulfilling the assumption of not interrupting the flow of the processed material. Based on inductive analyses, the concepts of improvement actions individually for each machine in the analysed subsystem were deductively defined. As a result, it was proved that a reactive maintenance strategy is appropriate for machines that have manufacturing reserves and are low priority. The equipment possessing manufacturing reserves but also having an impact on the risk of interrupting the flow of the processed material should be operated in accordance with a preventive maintenance strategy. A predictive maintenance strategy was proposed for the machines with the highest priority, which simultaneously do not have manufacturing reserves and the risk of manufacturing line operation interruption is high. The considerations were conducted with a holistic approach, taking into account the main functional areas of the enterprise.

**Keywords:** MTTF—mean time to failure; failure analysis; maintenance strategy of series-parallel system; reactive maintenance; preventive maintenance; predictive maintenance; make-to-order; system dichotomy

## 1. Introduction

Analysis consistent with the theory of complex systems [1,2] enables the decomposition of the entire system into sets of interdependent subsystems at different levels of the enterprise hierarchy (operational, tactical, strategic). Large enterprises (including manufacturing ones) with a complex organizational structure assess the effectiveness of the work of separate subsystems according to the hierarchy level at which they are located [3]. For each separated subsystem, the performance evaluation measure is adjusted [4]. The most popular method of assessing the effectiveness of the operational level tasks is using the following indicators: OEE—overall equipment effectiveness, OOE—overall operations effectiveness, and TEEP—total effective equipment performance [5]. The choice of one of the three depends on the organization of work in enterprises. The TEEP indicator is used in enterprises operating in a three-shift mode. Assessment consistent with OOE is used in sectors with a small number of relatively short changeovers. In turn, the OEE indicator

is most often used due to the assessment of the use of net time [6–8]. A review of the wide application of the OEE indicator along with an evaluation analysis is presented in the publication [6]. These indicators are the primary metric for total production maintenance (TPM) assessment [6] according to the Toyota Production System (TPS). TPS is primarily focused on maximizing the use of time. Therefore, the main evaluation parameter is time. The value of time units can be relatively easily converted into financial values [9].

The tactical level coherently transforms many operational level assessments that generate the final strategic level result. Hence, a frequent evaluation parameter is a financial indicator [10]. At the tactical level, the financial balance is also considered, including the energy costs of the company's operations, the costs of $CO_2$ emissions [8,11], currency exchange differences (for companies with an international reach) and the involvement of other resources that are necessary for the implementation of tasks (e.g., cooling water, etc.).

The ecological aspects of the enterprises functioning have various ranges and areas of operation [12]. From operational, through social, to global. At the operational level, efforts are made to minimize the negative impact of the manufacturing company's core business. This area generates the highest number of destructive factors for the local ecosystems. The type and quality of the materials used for manufacturing directly affect the amount of waste generated during production, as well as during the use of the product. This also applies to the technological equipment of the machinery park. Materials with better properties, optimally matched to the specifics of the enterprise's work, enable longer use without the need of replacing machine components. The low failure rate of machines in a strategic aspect means a longer service life. In the operational aspect, a high value of OEE, OOE or TEEP indicators is obtained by maximizing the use of working time. In the tactical aspect, it means a low resource involvement in spare parts and related processes [9,13]. Therefore, the optimization of maintenance rules has measurable effects [3,4], which translate into time gain [14], limitation of financial outlays [9,10] and reduction of waste in repair and maintenance processes.

Regardless of the type of analysed production system, the principles of maintenance of the machinery park play an important role in the strategy of tactical operation at the operational level. As a result of the expansion of the global economy, the maximization of financial efficiency is achieved with the use of highly efficient linear systems. The productivity of the series system is determined by the weakest link in the production chain. Therefore, in the field of maintenance, many scientific studies and publications are dedicated to the optimization of series systems [9,15,16] or analyses of a single machine [14,17–23]. Analyses of series-parallel systems are less common [24,25]. This is due to the dynamics of changes in the operating states of the system and the related level of complexity of the analyses [25,26]. Very often these systems for several evaluation parameters are NP-hard problems [20,21].

In ecological aspects, one of the main areas of assessment is the energy efficiency of production systems [27–29]. Ecologically conscious production companies strive for optimal use of resources. The assessment of energy efficiency directly reflects the level of wear of technical units (machinery). Therefore, the energy efficiency of various areas of the enterprise plays a significant role in the decision-making process [23], also from a strategic, long-term perspective. In the current political and economic situation, the low energy consumption of an enterprise is a determinant of its competitive advantage.

In the literature, differences in terms of nomenclature techniques and methods of machine maintenance are described. However, in a general division three different strategies can be distinguished: 1. reactive maintenance, 2. preventive maintenance and 3. predictive maintenance. Reactive maintenance and preventive maintenance are two main maintenance strategies defined in the standard EN 13306 [30,31]. In turn, predictive maintenance as a relatively new concept is developing as an alternative for preventive maintenance in order to ensure the increasing requirements of reliability, availability, maintainability and safety of systems with the use of testing techniques to monitor and evaluate equipment performance trends [32]. The popularised opportunity maintenance method is a hybrid approach of reactive and preventive maintenance in multi-element considerations. It should

be noted that the factor initiating repair actions in opportunity maintenance is the reaction to the failure [33]. The implementation of the remaining tasks is preventive actions that are performed incidentally as a result of stopping the main production tasks. Another combination of the two strategies is selective maintenance. This type of maintenance is focused on preventive actions, but as a result of limited resources, some of the maintenance tasks with a short replacement time remain in the reactive strategy [13] with the use of continuous diagnostics typical for the predictive strategy.

Reactive maintenance is a strategy where maintenance actions occur when a system or object fails. For this reason, the reactive maintenance model cannot optimise maintenance according to certain reliability or economic criteria [34,35]. Reactive maintenance is considered to be ineffective and expensive [36]. The article presents an example of a real production system in which for a series-parallel system it is justified to use a reactive strategy for individual units with technological and production reserves. In a holistic assessment of the system, considering the principles of sustainable development, it is advisable to take corrective actions in the event of undesirable random events (e.g., breakdowns, unplanned reduction of the work pace) but only for units with a low value of the priority number of devices (PND) [37].

Preventive maintenance is a strategy in which maintenance actions are undertaken to maintain a system or unit in a specific condition by performing regular inspections and detecting and preventing failures [34,38]. The main goal of preventive maintenance is the optimal planning of repair and overhaul activities. For complex production systems, the development of an optimal schedule of periodic inspections should consider the maximisation of the possible working time between failures (TBF) [39]. The time between failures is understood as the sum of the successive values of time to failure and time to repair. The strategy of frequent replacement of non-damaged components is against the general principles of sustainability. For high-performance production systems, higher economic efficiency is achieved by using a preventive strategy. Then, the existing functional and economic interdependencies are considered in the principles of sustainable development of the enterprise. Preventive replacement of the used components is an activity that reduces the risk of undesirable random events [40]. As a result, the continuity of the flow of processed materials is ensured throughout the entire supply chain.

Predictive maintenance is a concept based on the precise adjustment of activities that maximise the reliability of a technical unit. In predictive maintenance, we are able to accurately estimate the moment of failure [41]. The precision of estimation is the result of the analysis of a series of data relating to the object under consideration. Both in preventive and predictive maintenance, statistical analyses are of great importance in planning repairs and replacement of spare parts. For example, average times of correct operation between a specific type of failure $MTTF_i$ for $i = 1, 2, \ldots, I$; where $I$ determines the maximum possible number of different types of failures in a technical object. The accuracy of estimating the moment of $i$-th occurrence depends on the value of the dispersion in relation to the mean. For failures that are not the result of material fatigue of properly manufactured products, the moment of failure is a random event. Then the standard deviation $\sigma_{MTTF_i}$ takes relatively large values, and the determination of the time of the $i$-th failure is imprecise. In order to accurately determine the moment of failure, it is also necessary to collect additional information about the machine in addition to statistical data [42]. In order to perform the diagnostics of machines necessary to determine their technical condition and the necessity to perform maintenance, among others, thermovision, vibroacoustics, sensors for measuring acoustic emission, current consumption, etc. are used [43,44]. Collected data using MES (Manufacturing Execution System) class systems create a connection between real production and virtual space.

Integration of real material flow areas with virtual space creates networks of information flow connections with unlimited possibilities for their collection, processing and sharing. The fourth industrial revolution, referred to as Industry 4.0, covers a wide range of new technologies [45]. The main purpose of using the latest Internet of Things (IoT)

solutions in the area of a manufacturing company is to minimise operating costs while max­imising the effectiveness of activities [46]. For operational (production) level considerations, this translates into increased efficiency and productivity. The reliability of technical equip­ment has a direct impact on the achieved efficiency [47]. Hence, predicting and optimising machine life is currently one of the main areas of sustainable production [32,48,49].

The article analyses a complex manufacturing system with a high level of customi­sation of final products. In order to develop a strategy for the operation of individual machines, a maintenance analysis should be carried out, which takes into account the selected operating properties of the system [50]. The analysis included the determination of the interdependence between the reactive, preventive and predictive maintenance strategy and the impact of the selection of the strategy on the principles of sustainable development. In most cases, these analyses relate to linear systems [51–53] or single-machine analyses of mixed systems. The article analyses a complex, series-parallel system. The analyses were multi-stage, complex—first inductive and then deductive. The first stage of the analysis was to divide the production system, in accordance with the theory of complex systems [1,2], into separate subsystems with a technological similarity to the performed production tasks. Then, based on the analysis of the continuity of the flow of processed materials, one subsystem was selected that met the conditions of a complex system with series-parallel flows. The analysed example is a system showing dichotomies—there are both flows of the type *push* and *pull*. Moreover, in the subsystem separated for analyses, there is one linear flow (units: MP1, M1, MP2, M2—Figure 1, in Mechanical Department). Hence, the main goal is to show that the selection of the optimal maintenance strategy should be adjusted individually to the elementary unit, considering the relations between other machines. The optimal choice of a maintenance strategy is important to maintain the continuity of flows not only in the area of the considered company but also in supply chains—both on the entrance and exit side of the enterprise.

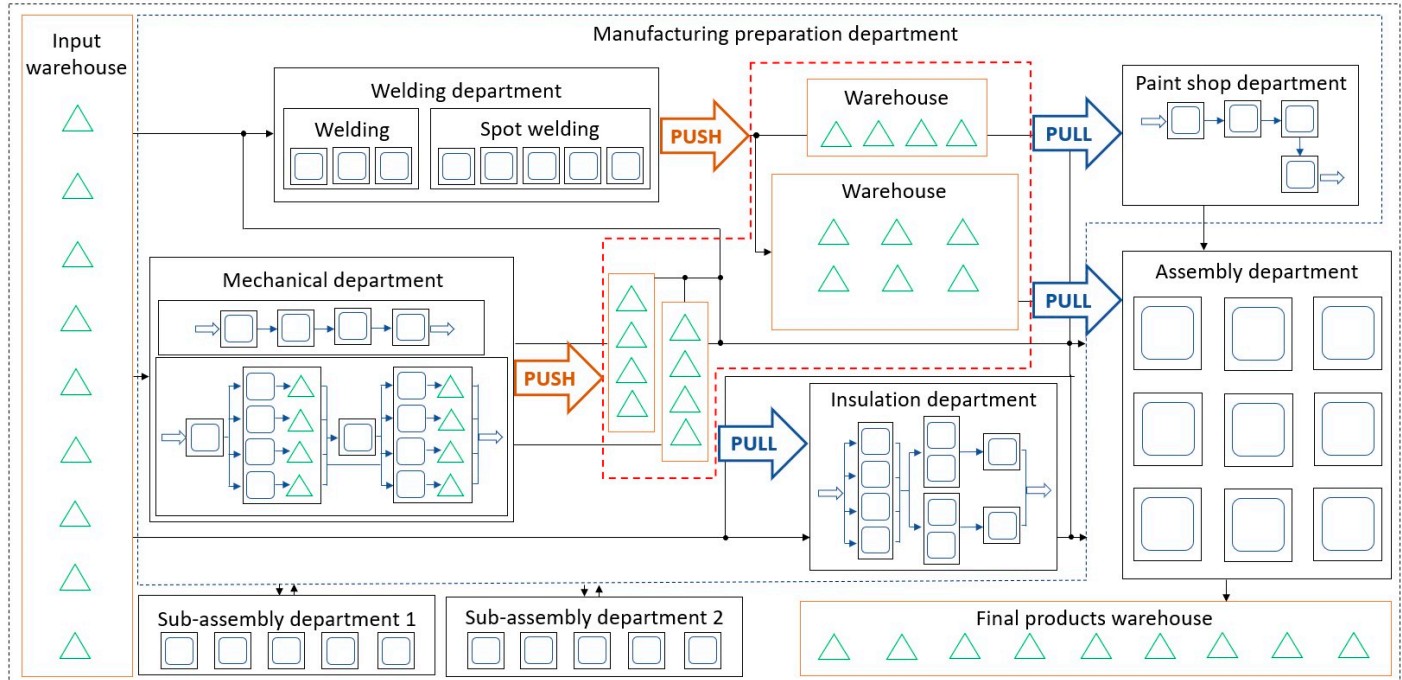

**Figure 1.** Scheme of separated subsystems of the serial-parallel manufacturing structure.

## 2. Materials and Methods

The description of the production system under consideration covers a multidimen­sional area of analysis: functional, structural and process. The analyses were performed in

two stages: first inductively and then deductively. The individual stages of the considerations were carried out in the following order:

- Analysis of the scope of customisation of the offered final products;
- Division of the enterprise according to the functional criterion: supply, production, distribution along with the definition of key parameters conditioning the operational activities of individual areas;
- Analysis of the production area, which included a multi-level decomposition of the system into a set of subsystems according to the theory of complex systems;
- Analysis of the continuity of flows of processed materials with the separation of *push* flows and *pull* flows;
- Development of an algorithm for determining the number of flows in a series-parallel structure that can occur at time *t*;
- For the subsystem adopted, the values of failure rates: MTTF—mean time to failure, MTTR—mean time to repair, MTBF—mean time between failures, were determined as well as the determination of the value of the dispersion in relation to the mean (σ);
- Developing a methodology for shaping the optimal TTF time value—time to failures using the expected value of a gamma distribution adjusted to the data;
- Matching the appropriate maintenance strategy (reactive, preventive, predictive) to individual technical units, taking into account technological and operational reserves.

The purpose of the analysis of the scope of customisation was to determine the flexibility of production. The high level of flexibility resulting from the production of MTO (make-to-order) is characteristic of the technological production structure (series-parallel system). Grouping the machines into sets of the same functionality allows you to reduce the risk of non-execution of orders because of a failure. MTO production is characterised by short production series and high volatility of demand [54]. In the analysed case, the longest production series was 12 pieces of the same product. Orders of single final products accounted for 57% of all orders. The considered example cannot be classified as prototype production because customers do not define the technical parameters of the ordered devices. The customer only individualises the order from the available range of possibilities. The selection criteria refer to, among others: length dimensions, several different operating ranges of the device and visual external features, including colour, glazing, number of shelves, etc.

One of the factors in minimising production costs is the production of the so-called large-scale and mass production. Examples of mass or large-scale production are, among others: the automotive sector, household appliances and audio/video devices, cigarette production, and the pharmaceutical sector. For these cases, each failure during the execution of a batch of a specific order may result in a lack of timeliness. Delay costs are very high in global supply chains. Hence, for the aforementioned industries with a wide distribution range, it is financially effective to use preventive or predictive maintenance [36,55–57].

Reducing the size of the production batch, while optimising the length of changeover times, has a two-way impact on the importance of the use of reactive maintenance. The first aspect is the minimisation of the risk of delayed deliveries resulting from the scale effect. With the appropriate organisation of production, a failure may result in delayed delivery of goods to customers with a small strategic share (the so-called choice of the 'less bad'). The second aspect is the diversification of the machine park resulting from the nest organisation of the production of the so-called technological structure. Due to the differentiation of the machines of technological similarity, we obtain the potential of technological and production reserves. This aspect is related to the analysis of the production area, including optimisation of the levels of labour consumption of works in the course of production, scheduling of tasks at the operational and tactical levels. These analyses will be discussed in detail later in this article.

The second stage of the analysis was to define the key parameters conditioning the work of individual functional areas. Analyses of primary demand, i.e., the distribution of final products (the so-called "enterprise outputs") concern the previously described scope

of customisation. Primary demand determines secondary demand, that is, the demand for materials on the "entry into the enterprise" side. The long-term activity of the company has developed an individual policy of cooperation with many suppliers. The relatively large variety of suppliers translates into different lead times for orders. The longest contracted order fulfilment time, during which no undesirable random events occurred, is 18 weeks. The length of this time directly translates into the order fulfilment time for the customer. The company's success over the competition is evidenced by the relatively short distribution time. On average, it is two to three weeks. In order to achieve such a short delivery time to the customer, with the lead time of the order many times higher, it is necessary to have high stock levels—in particular, stocks for products with the longest supply times. Having a high stock level (both on the supply and distribution side) is economically ineffective. However, for the enterprise under consideration, the short lead time for any customer order is a key competitive advantage. Hence, the functional analysis of individual areas of the enterprise referred to the time of the streams of deliveries to the customer and the delivery times of the ordered semi-finished products. The functional analysis also takes into account the transition time from entry to exit from the enterprise. However, in order to be able to separate the so-called critical path of the production lead time, it was first necessary to perform the third stage of considerations: decomposition of the system into sets of subsystems according to the general theory of complex systems [1,2].

The choice of the criterion of system decomposition into separate subsystems depends on the assumed research objective. In the discussed example, the aim is to determine the impact of the machine reliability maintenance strategy on the principles of sustainable development of the company. Hence, in most analyses, the main evaluation parameter is the duration of the event in question (e.g., delivery time, order time, process duration, repair duration, etc.).

The nest-based organisation of the production system consists of the grouping of machines implementing similar or the same technological processes. For the company discussed in the article, seven main subsystems with a different number of associated machines were separated (Figure 1). By making a subsequent division of the hierarchical structure of the production system in each separated subsystem, it is possible to distinguish subsequent subsystems of the superior system. For the purposes of the analyses in the area of the article, subsystems up to the third level of classification were distinguished. Figure 1 shows a simplified diagram of the separated subsystems of the analysed production structure.

The complexity of the flow analysis is the result of the number of possible planning routes for given production technology, the number of available machines and different types of material flow. In the analysed case, there are two different flow streams: *push* and *pull* (Figure 1). Push flows occur in the first stages of material processing; pull flows occur in the next part of the system. The existing dichotomy generates a higher level of work in the production of WIP (WIP—work in progress) but reduces the potential risk of stopping the continuity of the flow of processed materials. Such a situation was developed on the basis of many years of experience. In terms of minimising the level of inventories in the course of production, these activities are financially ineffective. However, in assessing the maximisation of the level of use of the available net time, it is a beneficial solution.

Analyses of the impact of the strategy of maintaining the reliability of machines (reactive, preventive or predictive) on the principles of sustainable development of the enterprise should take into account the number and type of flow streams of the processed material. The number of flow streams of the processed material results from the technical and technological solutions implemented in the company. For the nest organisation of production, the determinant of the number of flow streams of the processed material is the number of possible configurations of the planning route. For a given technological route, with the access of more than one machine implementing a given production process, we can define $M$ different planning routes. Graphic visualisation of the cited case is presented in Figure 2. In Figure 2, subsequent types of manufacturing processes in an example

series-parallel structure are marked as P1–P4 and the manufacturing units performing specific manufacturing processes are marked as 1.1–4.2.

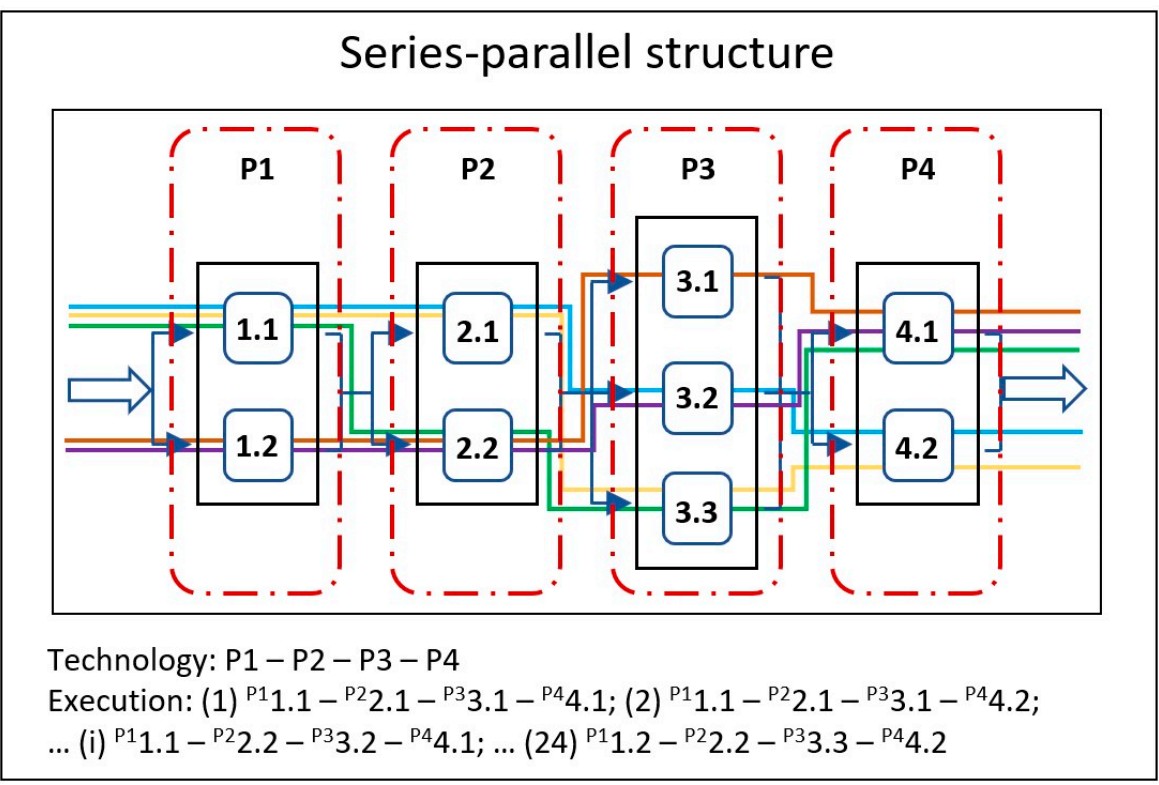

**Figure 2.** Graphical visualisation of possible flows of the same technological route.

The differentiation of the assortment of final products, short production series, and the complex BOM (bill-of-materials) structure determines the complexity of the system and the complexity of the analyses. In the area of the case under consideration, two parameters have an impact: the number of components and the complexity of the relationships between the system components. If we define the flow of the processed material with a specific planning route as a set of a sequence of relations, then the maximum number of relations will be determined according to the formula:

$$L_{AR} = n(n-1), \tag{1}$$

where: $L_{AR}$—maximum number of asymmetric relations, $n$—number of system or subsystem components (in the presented case it is the number of machines).

The article assumes that there are no feedback relations on any component of the system (machine), i.e., no machine is in relation to itself. In addition, the considerations relate to unidirectional flows (input $\rightarrow$ output), hence symmetrical relationships were also excluded.

In the reliability analyses, the machine has two readiness states: fit for work and unfit for work (it is in the state of failure, inspection, changeover, etc.). If the appropriate states mean: 1—the machine is working and 0—it is not working, then in state 1 there is an asynchronous relation between two dependent system elements. There is no such relation for state 0. Then, at time $t$, the state of an $n$-element system is defined by a sequence of zero-one numbers. The number of possible operating states of the system ($L_{SP}$) at any time $t$ can be determined according to the Equation (2):

$$L_{SP} = 2^{L_{AR}}, \tag{2}$$

where: $L_{AR}$—maximum number of asymmetric relations determined according to the Equation (1).

For example, for three separated subsystems there are two machines in each, according to Equation (2), the number of possible operating states of the system at time $t$ is $L_{SP} \cong 1 \times 10^9$. Enterprise-wide holistic layout analyses belong to the NP-hard task class. Hence, further analyses of the impact of the choice of the machine park reliability maintenance strategy on the sustainable development of the company were made for one subsystem. The mechanical department—MD, made up of the following technical units, was found to be the appropriate representative for the overall system:

- Automated sheet metal processing line (machines: MP1, M1, MP2, M2);
- Four hammer punches with different working fields (Table 1) (machines: CM1, CM2, CM3, CM4);
- Four working press brakes with a permissible bending length of 5 metres (machines: BM1, BM2, BM3, BM4).

The mechanical department (MD) also includes single sheet metal cutting machines (CMO) and a deburring machine (DM). These objects do not have a significant impact on the criteria of the conducted analyses. Only 3–4% of the components are subject to the deburring process. Moreover, both machines have low utilization rates (no more than 40%). Figure 3 shows a diagram of the analysed system with marked material flow streams. In Figure 3, triangular symbols mark the warehouses included in the production line. They are not production units; therefore, they are not subject to the analysis conducted in the article.

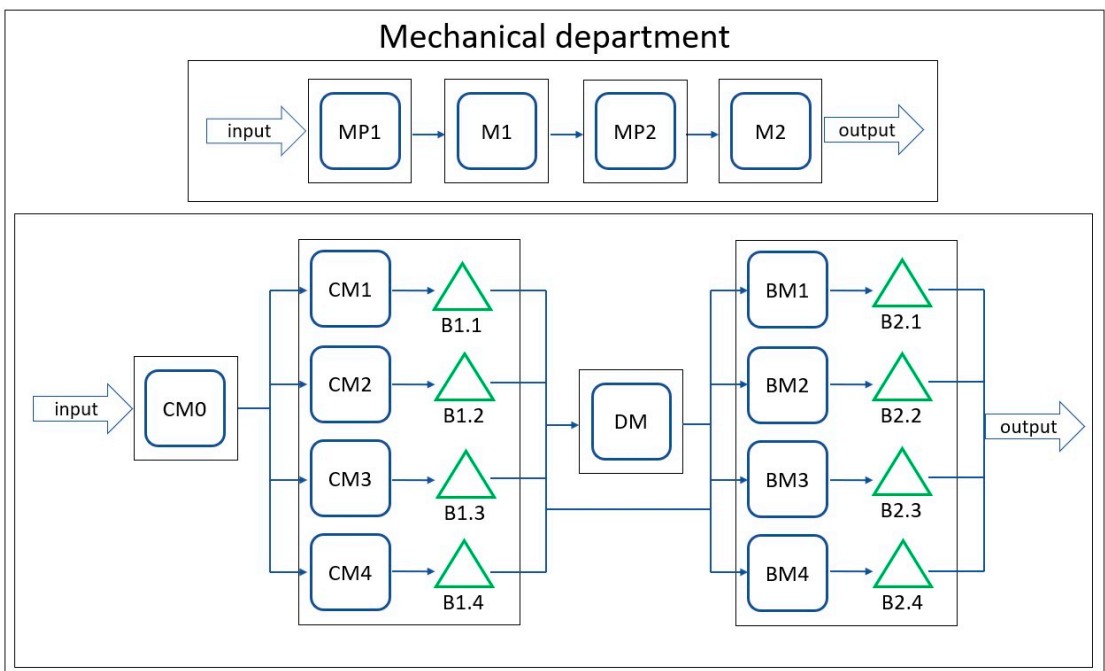

**Figure 3.** Scheme of the analysed structure—mechanical department (MD).

For the determined system, there are two different material flows: linear flow (route: MP1-M1-MP2-M2) and 32 possible technological flows (Figure 4). A characteristic feature of linear flows is the production of products in a strictly defined planning route with a consistent sequence of technological routes. In this system of material flows, all processes are highly interdependent. Any delay or stoppage of a single stage immediately results in a stoppage of the entire line. Hence, failures in this system cause large losses in productivity and are very costly in terms of the entire system. A failure of one module suspends the production of all planned components along the entire line.

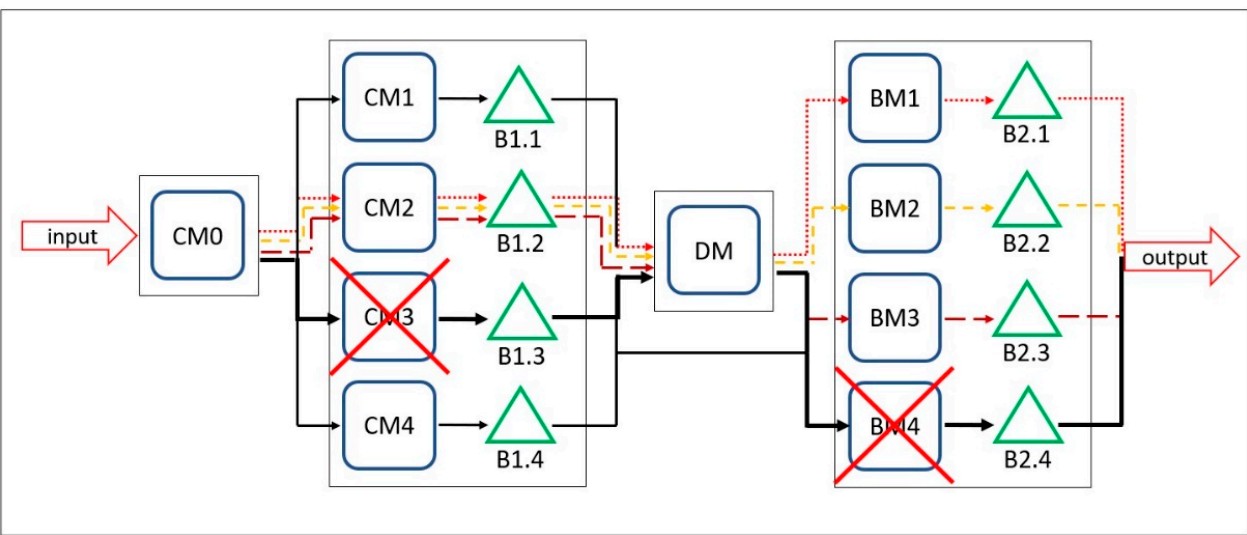

**Figure 4.** Examples (potential) of material flows in a series-parallel system.

The situation is different in mixed (series-parallel) systems. For parallel objects of technical similarity (sets: {CM1; CM2; CM3; CM4} and {BM1; BM2; BM3; BM4}—Figure 4) failure of one of the machines requires the development of a new production schedule for the other two parallel objects. This example is shown in Figure 4.

In order to be able to redirect production tasks from a machine that is in a failure state to another technical unit, the following conditions must be met:

- The possibility of implementing technologically identical tasks, and the result of which will be the same in terms of the quality of performance;
- There are reserves for the levels of the labour intensity of works in the course of production,
- The appropriate specialisation and the assumed level of quality of workmanship are ensured.

For machines belonging to the considered system, a list of reserves meeting the above-mentioned conditions was prepared (Table 1).

**Table 1.** Technological reserve of the production cells CM1, CM2, CM3, CM4.

| Machine Name | Working Area Dimensions (m × m) | Machine Reserves |
|:---:|:---:|:---:|
| CM1 | 1.5 × 1.5 | CM2/CM3/CM4 |
| CM2 | 2 × 2.5 | CM3/CM4 |
| CM3 | 2 × 2.5 | CM2/CM4 |
| CM4 | 2.5 × 4 | - |

The reserves of machines carrying out an identical technological process are defined according to the size of the working area. In the production line under consideration, four machines are used to carry out the hammer punching process: CM1, CM2, CM3 and CM4. They differ in the size of the working area (see Table 1 for a summary). In the event of a failure on the machine with the smallest working area (CM1), the punching process can be performed with any of the other three machines. In the event of a failure on one of two machines with the same size of the working area (CM2 or CM3), the processes can be redirected to one of the other two machines in working order with a working area of at least 2.5 metres. On the other hand, in the event of a breakdown of the machine with the largest working area (CM4), it is not possible to redirect the processes requiring the maximum working area to another machine. Due to the lack of a reserve, the CM4 machine has the highest PND value. The methodology of determining the value of the machine priority has been presented in publications [58–60].

The analysis of reserves in terms of technological possibilities and labour consumption levels showed that CM2, CM3 and CM4 machines are machines of high priority, of which the C4 machine is a critical object. The priority of machines numerically determines the importance of the machine in three evaluation criteria: the percentage level of use, technological reserves held and the impact on the continuity of production in further flow streams of the processed material. The considered system (Figure 4) is a separate subsystem from the entire superior system (company system), which is at the beginning of the production line (Figure 1). The production delay in the analysed system is each time determined by delays in the further part of the production system. It was assumed that the impact on the production continuity behind the system is very high. Depending on the rating scale (usually from 1 to 10), this value was the maximum for each machine. With these assumptions for a series-parallel system, the failure rate of machines affects the changes in the production schedule and the potential overload with too many orders to be performed. Systems showing high variability should have algorithms that control in real time the allocation of labour consumption levels in the course of production to each individual production station [53].

Statistical control of the failure rate of machines is used to estimate the moments of failure. The most common failure assessment indicators are: MTTR—mean time to repair, MTTF—mean time to failure and MTBF—mean time between failures [61,62]. The assessment according to the above-mentioned indicators can be made collectively, taking into account all types of failures. Then, the general technical condition of the unit is determined. The relatively low value of the MTTF indicator indicates frequent failure of the machine; this may result from the high level of wear and tear of the unit. In the article, for selected machines, a detailed assessment of the values of indicators for each of the *j*-th types of failure was made. Then the average values of the indicators are determined according to the formulas [62]:

- MTTR–mean time to repair

$$MTTR_j = \frac{\sum_{i(j)=1}^{N_{R(j)}} T_{R_{i(j)}}}{N_{R(j)}}, \tag{3}$$

where: $T_{R_{i(j)}}$—single repair duration of the *j*-th kind; $N_{R(j)}$—the maximum number of repairs of the *j*-th type of failure.

- MTTF–mean time to failure

$$MTTF_j = \frac{\sum_{i=1}^{N_{P(j)}} T_{P_{i(j)}}}{N_{P(j)}}, \tag{4}$$

where: $T_{P_{i(j)}}$—total duration of correct operation between the *j*-th type of failure; $N_{P(j)}$—the total number of correct operation events between the *j*-th type of failure.

- MTBF–mean time between failures

$$MTBF_j = MTTF_j + MTTR_j, \tag{5}$$

where: $MTBF_j$—a cumulative value of time between failures of the *j*-th type.

The accuracy of estimating the potential value of the correct operation time and the failure removal time directly depends on the spread in relation to the average value. An important parameter in the predictive assessment of machine failure rates is the value of standard deviation. In the analysed case, the value of the standard deviation was determined depending on the number of failures in individual units. For empirical data, the sample size of which is less than 30 elements, the value of the standard deviation was defined as the biased maximum likelihood estimator—Equation (6). Otherwise, when

the size of the empirical sample was more than 30 elements, the standard deviation was defined as the unbiased estimator of the maximum likelihood—Equation (7).

$$\sigma_{\dot{n}<30} = \sqrt{\frac{\sum_{i=1}^{\dot{n}}\left(t_i - \bar{t}\right)^2}{\dot{n}}}, \tag{6}$$

$$\sigma = \sqrt{\frac{\sum_{i=1}^{n}\left(t_i - \bar{t}\right)^2}{n-1}}, \tag{7}$$

where: $t_i$—$i$-th element of the empirical sample for $i = 1, 2, \ldots, \dot{n}$ or $t_j$—$j$-th element of the empirical sample for $j = 1, 2, \ldots, n$; $\bar{t}$—arithmetic mean of the sample.

First, the global values of MTTF, MTTR and MTBF indicators for each machine were calculated individually. The values of the dispersion in relation to the mean ($\sigma$) were also determined (Table 2).

**Table 2.** Global values of MTTF, MTTR, MTBF indicators with a value of $\sigma$.

| Production Unit | MTTF (min) $\sigma_{MTTF_{(machine)}}$ | MTTR (min) $\sigma_{MTTR_{(machine)}}$ | MTBF (min) $\sigma_{MTTB_{(machine)}}$ |
|---|---|---|---|
| MP1 | 2194.8 | 17.2 | 2212.0 |
| | 5044.9 | 14.0 | 5047.3 |
| M1 | 2216.5 | 20.7 | 2237.2 |
| | 4945.6 | 16.7 | 4945.1 |
| MP2 | 34,615.2 | 16.2 | 34,630.1 |
| | 56,556.8 | 9.5 | 56,555.6 |
| M2 | 21,066.6 | 12.2 | 21,078.4 |
| | 25,494.8 | 4.9 | 25,496.8 |
| CMO | 6281.4 | 13.9 | 6295.2 |
| | 13,823.8 | 10.4 | 13,825.6 |
| CM1 | 7418.7 | 13.5 | 7432.1 |
| | 14,359.5 | 9.9 | 14,360.6 |
| CM2 | 2361.7 | 21.9 | 2383.5 |
| | 4395.8 | 30.0 | 4395.3 |
| CM3 | 1327.7 | 19.7 | 1347.4 |
| | 2774.4 | 28.5 | 2775.4 |
| CM4 | 4002.5 | 14.1 | 4016.3 |
| | 6498.9 | 10.1 | 6500.7 |
| DM | 1332.7 | 20.4 | 1353.1 |
| | 2733.0 | 19.7 | 2732.2 |
| BM1 | 23,756.7 | 12.3 | 23,768.5 |
| | 23,227.4 | 6.9 | 23,225.0 |
| BM2 | 32,278.5 | 32.3 | 32,310.1 |
| | 32,983.9 | 44.9 | 32,970.6 |
| BM3 | 14,831.2 | 22.7 | 14,852.9 |
| | 26,687.8 | 19.5 | 26,683.4 |
| BM4 | 15,3014.0 | 14.0 | 153,022.7 |
| | 176,564.1 | 3.5 | 176,557.5 |

The analyses of the MTTF values and their dispersions against the mean have shown that for the case under consideration, the value of the standard deviation is greater than the average time of correct operation. Only for the BM1 machine is the value of the standard deviation slightly lower than the MTTF value. The visualization of the duration data $TTF_i$ presented in the form of a boxplot for CM1, CM2, CM3 and CM4 shows the number of outliers beyond 1.5 IQR (Figure 5).

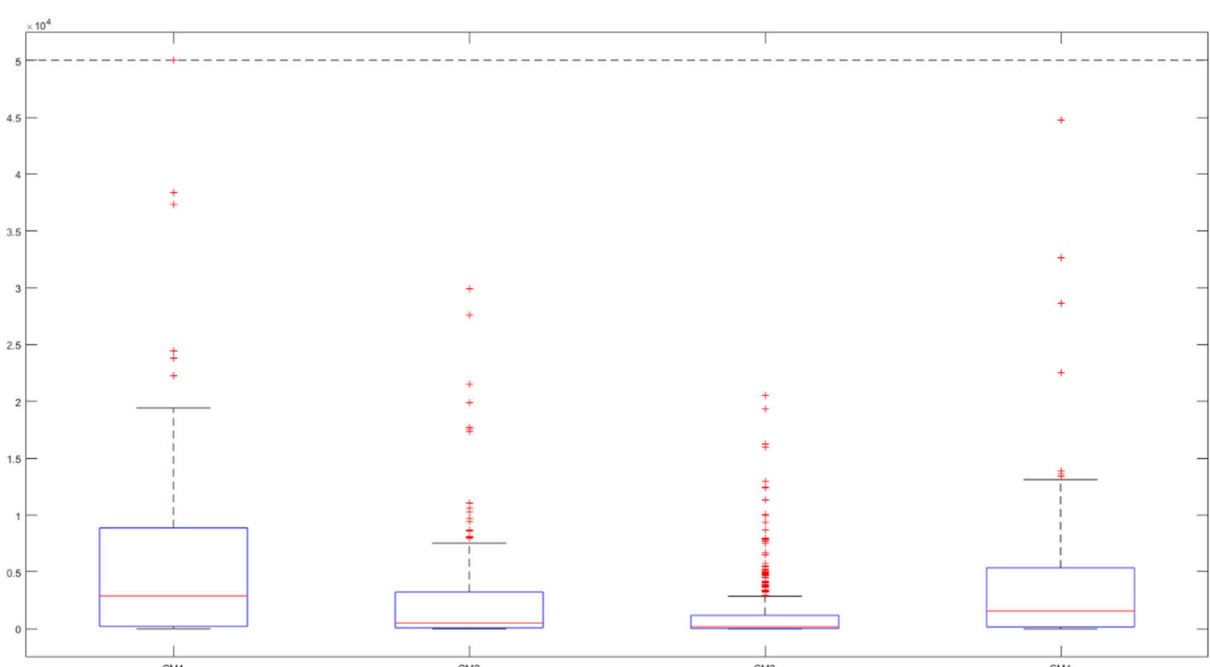

**Figure 5.** A boxplot of TTF (time to failure) values for units: CM1, CM2, CM3, CM4.

Such large randomness in the data shows that there are inconsistencies. A detailed explanation of the existing situation indicates an incorrect assignment of downtimes. In most cases, after a failure occurs, the machine starts up and the correctness of the process is verified. Then, the duration of the failure should refer to the sum of the failure and start-up time until the final start of the line. In the system under consideration, each time the machine is stopped, the operator is required to enter a reason for the stoppage. Hence, each stop is identified as a separate failure. From the operational level, it is not possible to cumulate machine start-up after failure as one TTR.

The analysis of failure rates according to Equations (3)–(5) refers to the CLT—central limit theorem. Predicting the times of correct work and failures in preventive maintenance, modelled with the normal distribution in industrial practice, is very often used. The utility of Equations (3)–(7) results from the easy estimation of the values of any random variables. In the case under consideration, the application of the normal distribution determines large estimation errors resulting from the spread of the data. Moreover, in the case under consideration, modelling with the Gaussian distribution, TTF (time to failure) and TBF (time between failure) times is burdened with a high probability of negative values. This situation is illustrated in Figure 6a,b.

In order to maximise the machine uptime, it is most advantageous to use a reactive maintenance strategy; this approach is also beneficial in terms of the principles of sustainable development. However, in order to maintain a balance between maximising the productivity of the production system and minimising the expenditure resulting from premature replacement of spare parts and repairs, a more accurate time estimation model was proposed. For this purpose, the gamma distributions with the parameters $\alpha$ and $\beta$ were adjusted to the approximation and modelling of the real times of correct operation and the duration of the failure. Using the method of moments, a system of equations was derived to determine the optimal values of the parameters of the gamma distribution (Equation (8)) [63]. For a small sample, the estimation of the distribution parameters is burdened with large fitting errors. Therefore, for the BM4 unit, the optimal parameters of the gamma distribution were not determined (the number of failures in the analysed period of time is $\dot{n} = 3$).

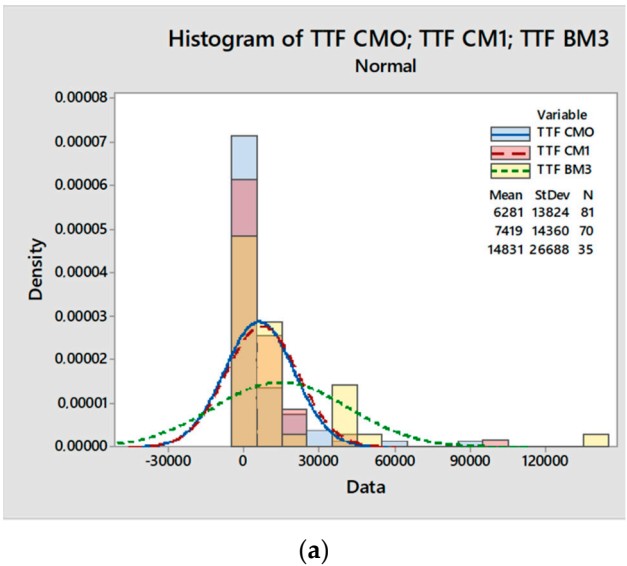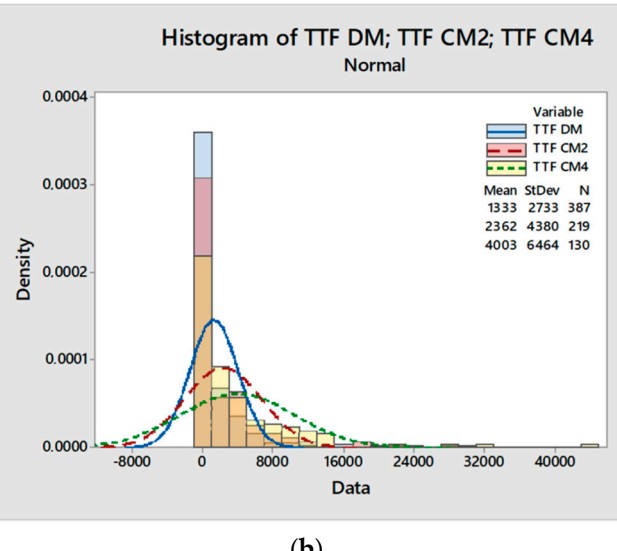

(**a**)  (**b**)

**Figure 6.** TTF (time to failure) frequency histograms and estimated Gaussian distribution diagram for machines: (**a**) CMO, CM1, BM3; (**b**) DM, CM2, CM4.

$$\begin{cases} \hat{\alpha} = \frac{4}{A_s^2} \\ \hat{\beta} = \frac{2}{s \cdot A_s} \end{cases}, \tag{8}$$

where: $\hat{\alpha}$—optimal value of the shape parameter of the approximated gamma distribution; $\hat{\beta}$—optimal value of the scale parameter of the approximated gamma distribution; $A_s$—skewness from the empirical test determined according to the Equation (9); $s$—standard deviation determined according to Equations (6) and (7).

$$A_s = \frac{\frac{1}{n} \sum_{i=1}^{n} \left(t_i - \bar{t}\right)^3}{\sqrt{\frac{1}{n} \sum_{i=1}^{n} \left(t_i - \bar{t}\right)^2}^3}, \tag{9}$$

where: $t_i$—$i$-th trial time TTF (time to failure) or TTR (time to repair); $\bar{t}$—arithmetic mean of the empirical test times.

Based on [63] the specified values of the parameters $\hat{\alpha}$ and $\hat{\beta}$ and the Equation (10), the expected value $E(t_{TTF})$ of the gamma distribution was determined.

$$E(t_{TTF}) = \frac{\hat{\alpha}}{\hat{\beta}}, \tag{10}$$

The determined values of the parameters $\hat{\alpha}$ and $\hat{\beta}$ and the expected value $E(t_{TTF})$ of the approximated gamma distributions for individual machines are summarised in Table 3. The table also includes the goodness of fit values $g$ of the gamma distribution determined according to the Equation (11).

$$g = \frac{E(t_{TTF}) - MTTF}{s_{TTF}}, \tag{11}$$

where: $MTTF$—arithmetical mean of time to failure TTF; $s_{TTF}$—standard deviation from the sample of TTF.

**Table 3.** Optimal parameters of the gamma distribution approximated to the data.

| Machine Name | $\hat{\alpha}$ | $\hat{\beta}$ | $E(t_{TTF})$ | $\mathcal{g}$ |
|---|---|---|---|---|
| MP1 | 0.082 | $5.672 \times 10^{-5}$ | 1443.5 | $-2.288$ |
| M1 | 0.304 | $1.114 \times 10^{-4}$ | 2724.8 | 1.566 |
| MP2 | 0.865 | $1.645 \times 10^{-5}$ | 52,611.0 | 1.147 |
| M2 | 1.731 | $5.161 \times 10^{-5}$ | 33,545.8 | 2.520 |
| CMO | 0.273 | $3.777 \times 10^{-5}$ | 7218.7 | 0.606 |
| CM1 | 0.186 | $3.002 \times 10^{-5}$ | 6189.4 | $-0.711$ |
| CM2 | 0.344 | $1.334 \times 10^{-4}$ | 2578.2 | 0.727 |
| CM3 | 0.310 | $2.008 \times 10^{-4}$ | 1545.7 | 1.545 |
| CM4 | 0.376 | $9.440 \times 10^{-5}$ | 3987.1 | $-0.027$ |
| DM | 0.281 | $1.941 \times 10^{-4}$ | 1449.9 | 0.842 |
| BM1 | 1.041 | $4.393 \times 10^{-5}$ | 23,701.5 | $-0.011$ |
| BM2 | 2.718 | $5.287 \times 10^{-5}$ | 51,409.7 | 1.940 |
| BM3 | 0.363 | $2.257 \times 10^{-5}$ | 16,077.0 | 0.272 |
| BM4 | - | - | - | - |

Figure 7a,b graphically show the strength of matching the determined gamma distributions to the empirical data collected for the BM1 and BM2 machines.

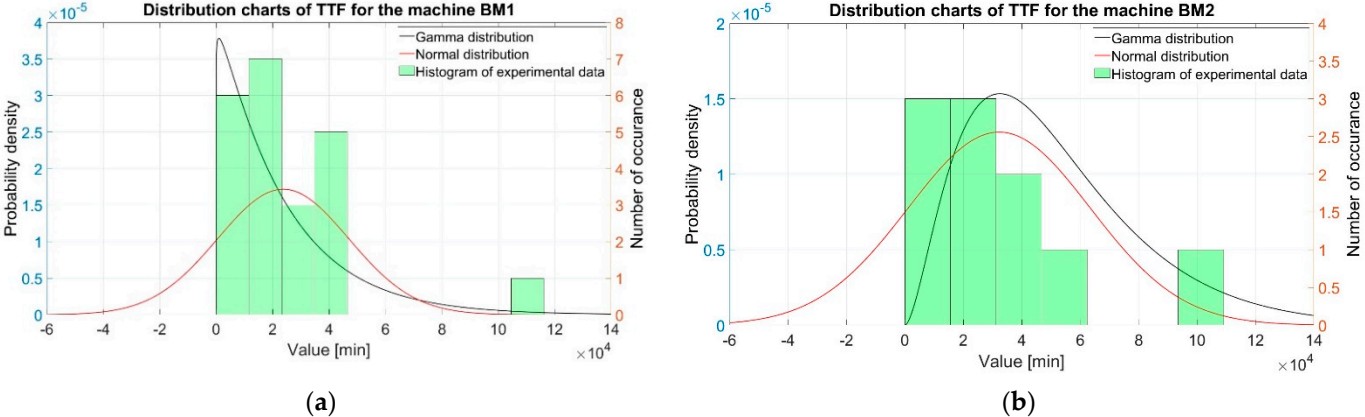

(**a**)            (**b**)

**Figure 7.** TTF (time to failure) frequency histograms and fitted normal and gamma distributions for machines (**a**) BM1; (**b**) BM2.

The closer the value of the goodness of fit index $\mathcal{g}$ is to zero, the better the approximated distribution is suited to the empirical data. The determined values of the strength of adjusting the gamma distribution for eleven cases oscillate around the value of 0 or 1 (Table 3). The remaining values are, respectively: $\mathcal{g}_{MP1} = 2.288$ i $\mathcal{g}_{M2} = 2.52$. The highest value of the force of fit was obtained for the BM4 machine BM4 ($\mathcal{g}_{BM4} \approx 8$). The number of empirical data for the BM4 machine is equal to $\dot{n} = 3$, at the same time the data have a large value of standard deviation ($\sigma_{MTTF} = 176\,564.1$). Hence, the estimation for a small sample and high randomness is always imprecise. The visual presentation of the gamma distribution fit, as well as the determined $\mathcal{g}$ values, prove that the prediction of the duration of correct work determined by the expected value of the gamma distribution is valid for the given case.

## 3. Results and Discussion

Holistic analyses of complex production systems are among the NP-hard tasks. The degree of difficulty of the analyses increases with the dynamics of changes in the operating states of the system. The class of systems taken into consideration is characterised by high dynamics of changes in operating states, which for a given $\Delta t$ can be determined according to Equation (2). The nest production structure is characteristic of unit production

or production with short production series. The smaller the number of elements in each individual series, the greater the system variability. In addition, systems in this class have a time-varying allocation of to-do tasks. All these factors have a significant influence on choosing the right maintenance strategy. In the case under consideration, the effectiveness of the applied maintenance strategy (reactive and preventive) was assessed, considering the principles of sustainable development. The main assumption was to adapt the type of strategy to a single machine, to make the most of the time between failures.

As a result of multi-stage analyses, the mechanical department subsystem was separated from the entire system in the first part. In the next stage, the streams of processed materials were identified to determine their rank of workmanship and the level of machine load. Then, the key values of the failure rates of machines were determined for the collected data on the duration of correct operation and failure times: MTTF—mean time to failure, MTTR—mean time to repair, MTBF—mean time between failures. In industrial practice, the failure rate of machines is most often determined according to the MTTF, MTTR and MTBF indicators. The analysis of the MTTF and MTBF indicators as a whole (without identifying the type of failure) informs about the wear status of the machine [62]. The analyses of failure rates according to Equations 3–5 refer to the central limit theorem (CLT) and the normal distribution associated with it. The collected data in the article adopted for consideration have certain inconsistencies in identification. Large values of the standard deviation (higher than the arithmetic mean—Table 2) showed that the adoption of the standard methodology of shaping predictive TTF times is burdened with a large prediction error. Therefore, an algorithm for determining the length of correct operation time according to its estimation was proposed as the expected value of the gamma distribution. The premise of the correctness of the application of the proposed method is the low value of the strength of matching the gamma distribution to the data. The smaller the value of the g index, the better fit the approximated distribution is. The value of the goodness of fit index for most machines is less than 2.5—Table 3.

The accuracy of the failure duration and repair duration estimation is important for ex-ante prediction. Another aspect of the assessment is the selection of a maintenance strategy in which the resource potential will be maximised. Machines strategically influencing the continuity of flows must be reliable; reliability is understood as operational certainty of operation [64]. Low-priority machines can operate under boundary conditions consistent with the principles of reactive maintenance. For these objects, the replacement of parts is not performed predictively (before the failure occurred). A different approach is recommended for objects with a high value on the priority index. Considering economic as well as ecological aspects, it is important to precisely determine the potential moment of failure. Therefore, for the estimation of TTF times, it was proposed to use the expected value of the gamma distribution (Equation (10)).

Shaping a maintenance strategy focused on the principles of sustainable development should consider the priority of the machine throughout the entire flow of the processed material. By focusing on a defined goal, determining the type of corrective action (reactive, preventive or predictive) depends on three main factors: utilisation level, potential reserves and the impact on the risk of delayed procurement. The operating state of the system, defined in a given $\Delta t$, also has a key parameter of the impact of the failure of a particular machine. In the subsystem separated for analyses, there are two technological similarity slots: first—a set of machines {CM1; CM2; CM3; CM4} and second—a set of machines {BM1; BM2; BM3; BM4}. Failure of a single machine for separate files determines changes in the value of the utilisation level on other machines in the given set. The assessment of the selection of a maintenance strategy is determined in the isolation system of dependence in relation to potential failures in other units. As a result of the conducted analyses, the following conclusions and observations were obtained:

- The DM and CM3 machines have the highest failure rate. In the analysed period, the number of unplanned stops was, respectively $n_{DM} = 387$ and $n_{CM3} = 388$. For both units, the cumulative failure duration constitutes 1.5% of the gross available time.

The DM machine does not pose a risk of stopping the flow of the processed material, because it is used for making up to 4% of the components. Hence, the optimal strategy for the DM machine is reactive.

- The CM3 machine has a higher failure rate, but it is an object for which there are reserves of two other machines: CM2 and CM4. Hence, the value of the level of risk of interrupting the continuity of flows as a result of the failure of the CM3 machine depends on the value of the accumulated time of production tasks of all machines in the set {CM1; CM2; CM3; CM4}. If the number of tasks necessary to be performed in time $\Delta t$ exceeds 90% of the maximum level of use, then a failure on any CM1-CM4 object is a determinant of delays in order execution. Such a situation may potentially occur in the period of the greatest demand for products with a high level of labour consumption. Hence, one of the stages of the research was to classify the final products according to the criterion of the accumulated time of task completion in each individual section of the system. Taking into account the fact that the system shows a dichotomy of flows (*push/pull*) the level of risk resulting from failure in the CM3 unit is low. On this basis, it can be unequivocally stated that for the CM3 unit it is necessary to use reactive maintenance.

- The CM1, CM2 and CM4 machines are units that implement the same technological process as the CM3 machine. Taking into account the fact that all three machines have a higher MTTF value in relation to the CM3 machine, reactive actions are recommended. Despite a large number of unplanned stops ($n_{CM1} = 70$; $n_{CM2} = 219$; $n_{CM4} = 130$)— but a low MTTR value and high unit capacity—all machines in the nest do not pose a potential risk of interrupting the continuity of the flow of processed materials. The uncertainty resulting from the high value of the standard deviation in relation to the MTTF will be reduced after the implementation of the proposed change in the method of identifying existing stops. This will also change the MTTR value (it will be larger).

- The CMO machine is the input to the subsystem under consideration. The machine has the highest unit capacity. The work of the machine takes place in one shift, compared to two shifts in other objects. Each stoppage (even the longest one) does not run the risk of stopping production. Proposed reactive maintenance actions.

- The task schedule of the entire analysed system is adapted to the machine nest {BM1; BM2; BM3; BM4}. These units determine the dichotomy of flows. Assuming the minimisation of the work inventory in the course of production, a large number of changeovers is necessary. The changeover time reduces the value of the net use time. Hence, the production schedule is focused on the accumulation of tasks that can be carried out on one changeover. This results in an increase in the share of the accumulated working time of the value added in relation to the gross time. Reactive maintenance rules will increase the risk of interrupting the continuity of the flow of processed materials. Preventive actions of the {BM1, BM2, BM3, BM4} units should be aimed at maintaining stocks of spare parts subject to breakdowns and having a resource of maintenance workers.

- The machines {BM1; BM2; BM3; BM4} are a few years old, hence their failure rate is very low ($n_{BM1} = 22$; $n_{BM2} = 10$; $n_{BM3} = 35$; $n_{BM4} = 3$). The number of failures and the low MTTR value directly translate into an almost zero share of failures in the cumulative gross time (BM1—0.05%; BM2—0.1%; BM3—0.15%; BM4—0.01%). The potential risk of production stoppage due to failure of one of the BM1, BM2, BM3 or BM4 machines is minor. The approach of the company's employees to date was determined by the terms of the guarantee. Currently, the service work schedule takes place regularly every quarter.

- A separate approach is the analysis of machines: MP1, M1, MP2, M2. These units work in a linear system. Failure of one of the machines of the linear system determines forced standstill of the others. The linear production system is typical for large-scale or mass production. In the case under consideration, on the MP1, M1, MP2, M2 machine line, standard components included in each final product are manufactured. Hence, a

predictive strategy is indicated [65]. Modern numerical processing systems, in their basic options, have the possibility of ongoing control of defined work parameters. In order to maximise predictive activities, it is advisable to implement algorithms for shaping the correlation between changes in operating parameters and occurring failures. Currently, the company has not identified the need to implement mathematical models to control for the correlations that occur.

In the era of the fourth industrial revolution, many scientific studies and publications mention the advantages of using solutions compatible with Industry 4.0. Maintenance 4.0 in its definition assumes predictive modelling of operational efficiency. It should be noted that this applies only to a few percent of production companies, mainly global corporations [44]. In the remaining production plants, the financial outlay for digitisation and virtualisation of the operating level is economically unprofitable [44].

One of the essences of the concept of the principles of sustainable development of the enterprise are ecologically justified activities enabling the economic development of the enterprise. With such assumptions, the condition of competitiveness in the market are actions within the boundary of economic and ecological constraints. The overriding function of optimising business activities is to maximise financial efficiency while meeting the conditions of ecological responsibility. Preventive replacement of spare parts generates more waste but reduces the risk of economic loss of profit. The ecological and economic balance is rarely used in relation to the maintenance strategy. Detailed analyses of the case under consideration showed that changing the organisation of accident data collection will increase the precision of TTF and TTR estimation. The accuracy of determining the moment of failure will allow the maximum use of the available working time without increasing the risk of delays in orders. In the long run, it will reduce the amount of waste of serviced parts and, as a result, reduce financial outlays.

It is common practice in real production systems to analyse empirical data according to the Gaussian distribution; this is mainly due to the accessibility and simplicity of analyses. In order to minimise the estimation error, the approximated distribution of the random variable should be appropriately selected. Hypothesis testing is a standard verification that the sample comes from a normal distribution. Data compliance with the normal distribution is used in the area of quality engineering. Machine failure rate estimation at the operational level is not a subject of such reliable analyses. The greater the simplifications, the greater the dynamics of changes in the operating states of the system.

## 4. Conclusions

In the TPM method, the values of correct operation duration are calculated as the arithmetic mean of all events. This method is sufficient for stable systems. The accuracy of time to failure (TTF) estimation is an important element in the optimal management of maintenance tasks. In the paper, an algorithm for determining the time of correct operation determined according to the expected value of the gamma distribution was proposed for a series-parallel system producing complex products in short production series (maximum 12 pieces). The determined values for the proposed algorithm were compared with the standard values. For the data showing a large dispersion in relation to the mean, the estimation according to the proposed solution shows smaller prediction errors.

In the next part of the research, a corrective action strategy was defined based on the level of machine utilization, operational reserves (for units in a parallel system) and the determined failure rates. The selection of reactive, preventive or predictive actions was considered elementary, i.e., for each machine individually. The originality of the conducted research is the holistic assessment of the complex system, where, on basis of inductive analyses, the concepts of corrective actions were individually determined. The class of systems adopted for analysis is characterised by high variability and the convergent nature of production in a system with a technological specification. In the first stage of the analysis, a complex system was decomposed into a set of subsystems according to the general theory of complex systems [1,2]. A representative subsystem consisting of fourteen machines was selected.

The assumed goal of the article was to propose an effective maintenance strategy, considering the principles of sustainable development.

The article emphasizes the problem of analytical solutions which, as Khairy and Kobbacy stated "are very unwieldy" in the implementation process [66]. This is due to the high dynamics of changes in the operating states of the system and the influence of many variables, which are difficult to study and describe individually. Therefore, in the area of scientific research, the application of complex heuristic, genetic or artificial intelligence (AI) algorithms is validated. The random nature of the processes causes a necessity to use appropriate models to calibrate the system evaluation parameters. One of the solutions is to use Bayesian inference algorithms [66].

The model for assessing the failure rate of machines presented in the article will be subject to the assessment of the accuracy of fit with the use of other distributions than the proposed gamma distribution in subsequent research works. The next area of analysis is the development of a Bayesian inference model. The predictive model will estimate the duration of correct operating for the time $\Delta t+1$, taking into account the evaluation of historical data in relation to the current operating states of the system.

**Author Contributions:** Conceptualization, B.Z. and J.W.; methodology, B.Z.; software, J.W.; validation, B.Z. and J.W.; formal analysis, J.W.; investigation, B.Z.; resources, B.Z. and J.W.; data curation, J.W.; writing—original draft preparation, B.Z.; writing—review and editing, B.Z. and J.W.; visualization, J.W.; supervision, B.Z.; project administration, B.Z.; funding acquisition, B.Z. All authors have read and agreed to the published version of the manuscript.

**Funding:** This research received no external funding.

**Institutional Review Board Statement:** Not applicable.

**Informed Consent Statement:** Not applicable.

**Data Availability Statement:** The data presented in this study are available on request from the corresponding author. The data are not publicly available due to the fact that the data are the property of the authors.

**Conflicts of Interest:** The authors declare no conflict of interest.

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
