# Peer review of "Selection of Maintenance Strategies for Machines in a Series-Parallel System"

_sustainability, doi:10.3390/su141911953_

Round 1

Reviewer 1 Report

50 TBF. Research results are missing.

235 Fig 2  P1, P2, P3, P4. 1.1, 2.1, 3.1, ... The text of the article does not disclose what these values ​​or indicators are.

244 ??? – maximum number of asymmetric relations,

257 ??? – number of asymmetric relations.

244 and 257 ??? not the same.

277 Fig 3  B1.1, B1.2, ... B2.1, B2.1,... The text of the article does not disclose what these values ​​or indicators are.

formulas (6) and (7) ?=1,2,…,?̇ or ?=1,2,…,?; ? not the same.

387 Fig. 5 There is a clear designation of the presence of an exponential distribution law of the theoretical law. But not normal or Gaussian.

419 Fig. 6 There is a clear designation of the presence of an lognormal distribution law of the theoretical law. But not normal or gamma.

574-614 The presented conclusions are more meaningful as a continuation of the discussion of the results obtained and contain elements of an annotation presentation.

Reviewer 2 Report

In this work, a series-parallel structure is studied and based on optimal MTTF maintenance-related decisions are made. In general, the authors have extended previous methods for complex systems and created a wider view in this field. Choosing different maintenance strategies for complex systems is an essential issue in this field, and articles can fill this gap. Some sections must be rewritten, and major revisions are needed.

 The article has been written carefully and in clear language. But The literature Review of this paper is poor and authors should add new related researches from reputable journals to  it. after literature review, the authors should clearly explain the main contribution of the paper.

The findings presented in the article can be clearly found in the literature review. For example, in lines 20 to 26, mentioning that corrective maintenance is appropriate for low-priority systems seems obvious. Even in line 43, the authors have referred to article 4 as a paper that has already achieved this issue.

Many works have been done in order to select optimal maintenance strategies for complicated-structure systems that each has beneficial advantages (for example: Tangbin Xia and others 2022, Optimal selective maintenance scheduling for series–parallel systems based on energy efficiency optimization), though the strategies used in this article have not been used in articles related to choosing the right strategy for complex systems, it is highly recommended to make a comparison with at least one article that have used different strategies such as articles based on energy efficiency optimization or based on cost minimization. In this way, the advantage of this article will be evaluated.

In the introduction, issues have been raised that have not been examined in any way in this article. Of course, if you enter them, the level of the article will be improved much more. For example, the use of modern technologies in the field of maintenance, IOT. There are also many irrelevant claims without proof and even reference, which must be removed. For example, in line 116, it is stated that it may reduce financial losses or in line 108, it is stated that it is better to have strategies aimed at reducing the negative impact on the environment, which have not been investigated in the article.

As mentioned in this article, the ecological effects are measurable. It is suggested that in a case study, the economic aspects and ecological effects for the proposed model be studied and the environmental and financial advantages should also be investigated.

Reviewer 3 Report

Journal: Sustainability

Comments on the manuscript entitled “Title: Selection of maintenance strategies for machines in a series-parallel system” (Manuscript No. sustainability-1855503)

The article is not suitable for publication in its present form. It needs a minorrevision. Below are my comments:

Some specific comments:

v  Abstract:  It is suggested to add some background with few objectives and possible applications of this study and highlight the novelty of this work. The abstract only contains some parameters without any process conditions or key values from the results, which is insufficient to delineate the whole pictures of contribution and possible application of this study.

v  Revise keywords add more specific and novel keywords with broader meanings (5-7 words).

v  Add some references in the Introduction section to strengthen the literature review

o   Hussain SZ, Kausar Z, Koreshi ZU, Sheikh SR, Rehman HZ, Yaqoob H, et al. Feedback Control of Melt Pool Area in Selective Laser Melting Additive Manufacturing Process. Processes 2021;9. https://doi.org/10.3390/pr9091547.

o   Kausar Z, Shah MF, Masood Z, Rehman HZ, Khaydarov S, Saeed MT, et al. Energy Efficient Parallel Configuration Based Six Degree of Freedom Machining Bed. Energies (Basel) 2021;14:2642. https://doi.org/10.3390/en14092642.

v  The introduction is lacks sufficient background information, which is unable to give the reader detailed background knowledge and possible wide application of this study. Research gaps should be highlighted more clearly and future applications of this study should be added.

v  Why you did only select this maintenance strategy (reactive, preventive or predictive), give justification with strong references.

v  Cite proper references to justify the equations used in this study.

v  Most important: Authors should prove their scientific originality by defining what are their main scientific findings, which have not yet been presented in other studies.

v  The tables/figures inserted are not explained or discussed well in the text please discuss critically and explain all tables/figures in the text.

v  No error bars were put in the figures. Shows the uncertainties in the data.

v  Conclusions of this manuscript lack clear findings and future aspects. The authors are advised to write the conclusion comprehensively.

v  Validate the results section with variables with the previously published results.

v  Between the lack of methodology and limited novelty, the authors have a significant amount of work to do to bring this paper up to a publishable standard.

Round 2

Reviewer 1 Report

The article has been significantly revised.

Reviewer 2 Report

The article is suitable for publication in its present form.